# Triggers and Barriers to Insight Generation in Personal Visualizations

Poorna Talkad Sukumar*
Union College

Anind Dey†
University of Washington

Gloria Mark‡
University of California, Irvine

Ronald Metoyer§
University of Notre Dame

Aaron Striegel¶
University of Notre Dame

## ABSTRACT

We report on the findings from a mixed-methods user study that explores some of the less-studied challenges in designing personal visualizations. We implemented an interface presenting visualizations of the personal data gathered as part of a prior study and conducted a think-aloud study (N=15) of participants' exploration of their respective data on the interface. We analyzed participant verbal reports and interactions to (i) corroborate the types of insights they gained with the insight types identified in the literature, (ii) identify contextual information recalled by the participants to interpret their data, and (iii) identify interface design choices that potentially hinder insight discovery. Our findings complement prior work and we present design directions for visualizations of personal data, including guidelines for providing baselines for comparison, providing higher explorability for more data-savvy users, and minimizing the need for reading even in more exploratory interfaces.

**Index Terms:** Human-centered computing—Visualization—Empirical studies in visualization

## 1 INTRODUCTION

Personal data comprises of any data that is relevant to one's personal life, such as, health and fitness data and social media interactions. Extensive prior work in personal informatics has studied various tools and aspects concerning the collection, presentation, and reflection of such personal data [15, 20, 26, 29]. Our work in this paper complements prior work and also differs in three regards. First, although visualizations are key to the presentation and reflection of personal data, the study of personal data visualizations and challenges in designing such visualizations within the InfoVis domain is relatively new and under-explored [24, 40] and we focus on exploring some of these challenges. Second, while prior work has mainly targeted *quantified selfers* or users who are highly motivated and familiar with tracking and interpreting their data [13, 15, 26, 29], we study participants with varied levels of interest and experiences with personal data and/or visualizations. Third, we present a unique study combining both reflection of *long-term* personal data as well as reflection of *historical* personal data, that have both been identified as being increasingly-common self-tracking practices [19, 47].

Personal visualizations refer to interactive visual representations of personal data and such visualizations can enable people to better understand themselves, share personal insights with others, and make changes to their behaviors [24]. There are, however, key challenges in designing personal visualizations that are less explored from a research perspective, such as, defining appropriate baselines

*e-mail: talkadsp@union.edu
†e-mail: anind@uw.edu
‡e-mail: gmark@uci.edu
§e-mail: rmetoyer@nd.edu
¶e-mail: striegel@nd.edu

for comparison and contextual information for facilitating interpretation [24]. This is due, in part, to the difficulty in studying personal visualizations and the need to use methods that assess their distinctive goals and study users interacting with their own data and in realistic settings [48].

Our approach to study personal visualizations involves leveraging the personal data gathered as part of a large-scale, longitudinal sensing study [34]. While such sensing studies, gathering diverse types of personal data (such as, sleep and physical activity), are generally aimed at studying individual and collective human behavior, they also typically include a component where participants are presented with their respective data gathered during the study [16], thereby providing ample opportunities for designing and studying personal visualizations.

The *Tesserae* study [34] ran between January 2018 to April 2019 and various personal data attributes were tracked from 757 information workers (across five cohorts/organizations) using wearables and Bluetooth beacons for one year. In collaboration with the researchers of the Tesserae study, we built a web-based interface (Fig 1) to present interactive visualizations of the personal data gathered from participants in the study. We conducted a virtual think-aloud study (N=15) with this interface to answer the following questions:

- **RQ1: What are the insights that participants draw from visualizations of their personal data and how do these insights compare with the personal-insight types identified in the literature?**

- **RQ2: What are the types of contextual information recalled by participants for interpretation of the data?**

- **RQ3: Which interface design choices are potentially hindering insight generation?**

Insight-based evaluations, where a visualization application's ability to support insight generation is evaluated by studying the insights gained by users [45], is a common method to evaluate personal visualizations [13]. Hence, through **RQ1**, we aim to validate our interface by studying the insights gained by participants. While prior work already identifies various types of personal insights [13, 14], we aim to corroborate these insight types through our study since the insight types can potentially vary across applications depending on the types of visualizations being used and context being studied (e.g., data exploration [14] vs. data presentation [13]).

**RQ2** and **RQ3** are motivated by the design challenges in personal visualizations identified by Huang et al. [24]. Given that quantified selfers routinely review their past or historical data on self-tracking tools [13, 19], we were interested in characterizing the types of context information recalled by participants to interpret their past data (**RQ2**). We conducted the think-aloud study from April to June 2020, more than a year after the Tesserae study ended and hence, we were well-placed to study *retrospection* in personal visualizations [19] and the types of contextual information recalled by participants. Additionally, given that the Tesserae study included participants with limited experiences with visualizations and/or data, we were

also interested in identifying interface design factors that present potential barriers to participants' effective interpretations of their data (**RQ3**).

The contributions of this work are the following: (1) a characterization of the types of context information that can be embedded within personal visualizations to trigger people's memory; (2) distinct exploratory behaviors of more data-savvy participants; and (3) design directions for visualizations in wearable and self-tracking apps, including guidelines for providing baselines, considering aesthetics, and presenting clear data and limited text.

## 2 RELATED WORK

### 2.1 Personal Informatics and Personal Visualizations

Personal data can refer to any data that is relevant to one's personal life [24]; examples include data from wearable devices (e.g., Fitbit), data pertaining to meetings and calendar events [4, 5, 25], learning progress [43], and Facebook interactions [49]. Personal informatics is a subfield of human-computer interaction that encompasses the entire spectrum of all activities and systems concerning the collection and reflection of personal data [15, 29]. Personal informatics research is especially inspired by the *quantified self* movement [2] empowering people to gather and reflect on personally relevant information through low-cost sensors and wearables [21, 26, 29].

Li et al. define personal-informatics systems as consisting of five stages – preparation, collection, integration, reflection, and action [29]. The *reflection* stage is where people self-reflect on their collected personal data and this reflection is mainly achieved by exploring and interacting with information visualizations. Hence, there is a specialized field of "personal visualizations" emerging within the InfoVis domain that focuses specifically on interactive visualizations of personally relevant information, the challenges in designing such visualizations, and suitable methods for evaluating such visualizations [24, 28, 48].

Personal visualizations have distinctive goals and usage characteristics; they aim to support self-reflection and provide actionable insights, are consumed in less formal contexts and often on mobile devices, and by people with different motivations and interests [3, 24, 27–29, 40, 46]. These distinct characteristics give rise to a number of challenges for designing and evaluating personal visualizations. Huang et al. present a survey on the use of visualizations and visual tools for personal data existing across many disciplines, including human-computer interaction and personal informatics, and discuss design challenges, such as, defining appropriate baselines for comparison and contextual information to facilitate recall in personal visualizations [24]. Additionally, studying personal visualizations requires the use of research methods that assess their distinctive goals and study users interacting with their own data and in realistic settings [48].

Empirical studies in personal visualizations have focused on evaluating their goals of supporting insight generation and behavior change and have predominantly employed qualitative research methods [26]. For example, Choe et al. characterize the types of personal insights gained by users by analyzing quantified selfers' video presentations [13]. Epstein et al. identify participant insights and opportunities for behavior change through an interview-based evaluation of their personal-visualization application, *Moves* [20]. In this paper, we supplement our prior work where we analyzed interaction logs to characterize the exploratory behaviors on a personal visualization interface [46] by conducting an insight-based evaluation of the same interface. In addition to assessing our interface's ability to generate insights, we also focus on some of the less-studied design challenges in personal visualizations [24].

Insight-based evaluation is a common method for evaluating personal visualizations. Choe et al. [13] define "insight" in the context of personal visualizations and present a set of personal-insight types by analyzing quantified selfers' video presentations. These insight types are further refined by Choe et al. [14] through a think-aloud evaluation of their personal visualization application, *Visualized Self*. We build on this prior work to identify the insight types gained by participants through our interface.

### 2.2 Think-aloud Studies for Evaluating Visualizations

The think-aloud method is a specific type of the observation method [10]. In the think-aloud method, participants are instructed to "think-aloud" when performing a certain task and verbal reports of their thought processes as they occur are captured and these reports are known to reflect conscious thought more accurately [12].

The think-aloud method has a diverse usage within visualization research. For example, think-aloud has been employed to glean user mental models or internal representations [32]. The method is also used in insight-based evaluations [13, 14, 37, 45]. The think-aloud method is also used in combination with other sources of data, such as interaction logs and eye-tracking data, to study exploratory behaviors with visualizations [7, 22, 52]. The method is also used to understand usability problems and improve the design of visualizations [9, 33]. In this work, we employ the think-aloud method for both insight-based evaluation and to understand usability problems.

## 3 DESIGN OF *The Tesserae Personal Data Explorer*

### 3.1 The Tesserae Study Design

The Tesserae study was a multi-university collaborative study concerned with studying aspects pertaining to information workers, including job performance, psychological traits, and physical characteristics [34]. Data was collected from 757 participants (across five cohorts/organizations) in complex information work professions for a period of one year. Participants were each provided with a Garmin vívosmart 3 wearable, which they were required to wear at all times during the study (except when charging the wearable). The wearable captured various health and activity metrics, including, sleep, steps, heart rate, and physical activity. Participants were also required to place two Gimbal Series 21 Bluetooth beacons–one in their homes and one in their workspaces and also carry one smaller Gimbal Series 10 key fob in their bags or key chains. They also had to install an app on their phones which captured both the data streamed from the wearable and sightings of the beacons whenever the phone came into proximity of the beacons. The beacon sightings were processed to determine the number of hours a participant spent at their home or in their workspace. In addition, participants were also required to answer various questionnaires as part of the study but we do not use the questionnaire responses as part of the data presented on our interface.

### 3.2 Design Decisions

We present below the design decisions for our personal visualization interface. Figure 1 shows a screen capture of our interface.

#### 3.2.1 Data

For each participant, we present six types of features and two types of summaries for each feature–(i) weekly summaries (or the daily averages for the corresponding feature aggregated over each week the participant was in the study) and (ii) monthly averages (or the daily averages for the corresponding feature aggregated over each of the 12 months the participant spent in the study). The six different features presented (in order) on the interface are activity, sleep, heart-rate variability (HRV), hours spent at home (home hours), hours spent at office (office hours), and number of steps (steps).

Activity, sleep, and steps data were obtained from the Garmin wearable that the participants wore during the study. Home hours and office hours were computed from the beacon sightings. We chose to present these features because they included both familiar features found on self-tracking apps (sleep, activity, and steps) as well as new features that were computed using the data gathered

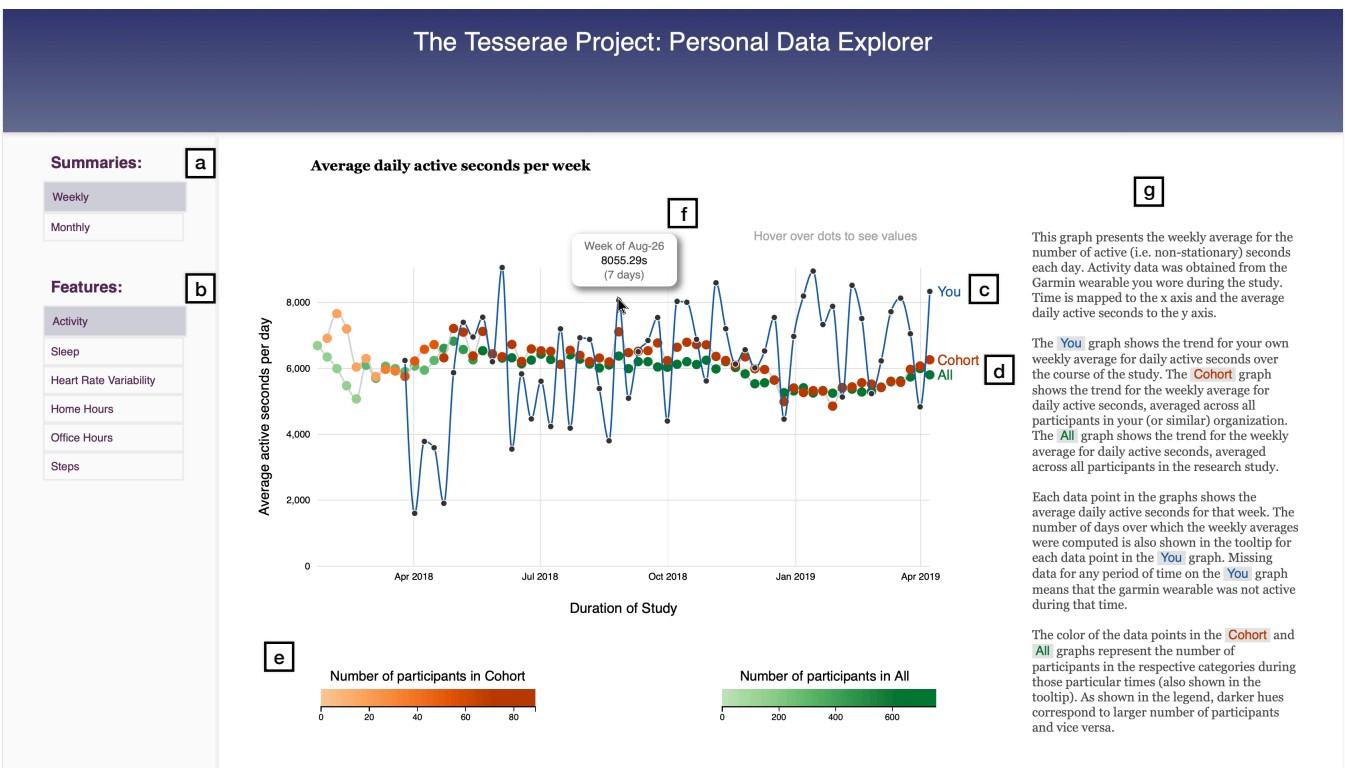

Figure 1: A screen capture of the personal visualization interface: (a) summary types, (b) feature types, (c) *You_graph* with participant's data, (d) *Baseline_graphs* with aggregate data of "Cohort" and "All" participants, (e) legend for "number of participants" within data points on baseline graphs, (f) tooltip with additional information displayed on hovering over data point, and (g) textual description of data and visualization.

from the Tesserae study (home hours, office hours, and HRV). Following recommendations from prior work for evaluating personal informatics systems [26], we designed our interface to be simple because it can be difficult to disentangle what aspects of the system lead to what findings or insights when evaluating complex systems. Hence, we did not include more complex data, such as, correlations between the different tracked variables.

HRV, indicating a measure of the time variability between each heartbeat [18], was computed using the beat-to-beat intervals (BBI) streamed from the Garmin wearable to the phone app. The SDNN technique (that is, standard deviation of the N-N, or beat-to-beat interval) [18], which computes the standard deviation in times between consecutive heart beats was used to compute the HRV values for the participants. Counter to heart rate, the interpretation of HRV is that a lower HRV value is indicative of feeling more stressed, while a higher HRV value is indicative of feeling more relaxed [18].

Participants had varying start and end dates and the amount of data visualized for a participant depended upon their data compliance in the study. We also included information on the number of days over which each (weekly and monthly) average was computed; in other words, how many days in the corresponding week or month the participant had data available.

### 3.2.2 Baselines

The Tesserae study provided us with opportunities to include baselines for comparison on the interface, one of the key challenges identified in designing personal visualizations [24]. We defined two baselines (which were included for each participant and each feature type)–the (weekly and monthly) aggregated data corresponding to the participant's *cohort* or organization as well as *all* the participants in the study.

### 3.2.3 Visual Encodings

All the feature types represent time-series data and hence, we visualized them using line graphs. Self-tracking data typically include a time component [14] and are commonly visualized using line graphs [15]. Hence, many of the insight types found in personal data contexts, such as, identifying values, identifying extremes, comparing values, and finding trends over time [13, 14], also happen to be insights associated with line graphs [51]. The line graph corresponding to the participant's data is labeled as "You" and the line graphs corresponding to the cohort and all data are labeled as "Cohort" and "All", respectively.

Given that the number of participants in the "Cohort" and "All" groups varied during the course of the study, to enable informed comparisons, we also visually encoded the number of participants in the respective groups by coloring the data points using sequential color scales.

### 3.2.4 Interaction Mechanisms

Participants could select between the two types of summaries (weekly vs. monthly) and among the six feature types. Participants could hover over each data point in the "You" graph to see details in a tooltip including the *x* value (week or month information, e.g., "Week of Apr-29") and *y* value (measured feature value with unit), and the number of days (of data included in the weekly/monthly average). Participants could hover over each data point in the "Cohort" and "All" graphs to view the *x* and *y* values and the number of participants in the respective group (cohort or all) during the corresponding time period.

### 3.2.5 Explanatory Text

In addition to graph titles, axes labels, and legend, we included explanatory text alongside each type of visualization describing (i)

the data shown, the $x$ and $y$ labels, and how the data were computed (that is, using the Garmin wearable or beacons), (ii) the three line graphs on the interface ("You", "Cohort", and "All") and what data they correspond to, (iii) what each data point represents in the "You" graph, the information displayed when they are hovered over, and what missing data points potentially mean, and (iv) what the color of the "Cohort" and "All" data points represent and the information displayed when these data points are hovered over.

## 3.3 Implementation

The visualization interface was developed using D3.js [1] and the same template was used for all the participants. We generated a unique, anonymized link for each participant which populates the data and baselines corresponding to the participant on the interface. We logged all the hover and click interactions of participants on the interface. We used AJAX to send details of the mouse/touch events as JSON objects to a PHP script which then logged them on the server.

## 4 THINK-ALOUD STUDY

### 4.1 Participants

The participants for the think-aloud study had all participated in the Tesserae study and were all from the same cohort or organization. We recruited participants from this cohort because these participants were not part of our earlier interaction-log-based study [46]. We initially planned to conduct an in-lab study. However, given the changed work conditions due to COVID-19, we then conducted the think-aloud study entirely remotely through *Zoom*[1]. We sent a "call for study participation" email to 84 participants from this cohort who had consented to be contacted for follow-up studies when they participated in the Tesserae study. We recruited the first 20 participants who expressed an interest in participating.

We initially planned to use the interaction logs (mouse clicks and hovers) automatically collected through our interface implementation to supplement the audio recorded during the think-aloud sessions. However, after conducting the study with five participants, we decided to also record the screen for the subsequent participants. This is because think-aloud data generally tend to be incomplete [12]; for example, when communicating insights, participants may point to data points on the visualization without actually identifying the data points out loud. Such collected think-aloud data can be more accurately and conveniently coded with the help of screen-capture recordings. We only use the audio, screen-captures, and interaction-log data gathered from the latter 15 participants in this paper.

Five of the 15 participants were female and 10 were male. All participants were full-time employees at [redacted for review] at the time of the study and worked in information processing domains. Two participants were 25–34 years old, six were 35–44 years old, five were 45-54 years old, and two were 55–64 years old. All participants had normal or corrected-to-normal vision and none were color blind.

### 4.2 Procedure

The first author conducted all the participant study sessions through Zoom on their 15-inch MacBook Pro. One-hour sessions were scheduled with each of the participants and they were sent instructions to use a laptop or desktop machine with reliable Wi-Fi access and with Zoom installed. They were also told that they would be required to have their video turned on and share their screen during the study.

At the beginning of each session, the participants were emailed the consent form and they were asked to share their screen as they read and signed the form. They were then asked to fill out a short questionnaire (with screen-sharing off) on their use of wearables and

associated apps after the Tesserae study ended, and their familiarity with reading visualizations in general, including the following question which we use as a measure of their **Data Savviness**:

- *How frequently do you browse your data on the mobile app(s) associated with these wearable(s)?*
  [1 (Never) to 4 (Almost all the time)]

Participants were then sent a unique link to the visualization interface populated with their respective data through the chat window on Zoom. They were asked to open the link in a browser window and instructed to freely explore the interface and to talk about their thought process as they did so. We employed free exploration because we wanted to study the insights gained by participants from freely exploring the interface as opposed to insights gained from performing specific tasks [45]. The participants were seeing the visualizations for the first time and the researcher did not provide participants with any explanation of the visualizations. The participants were required to share their screens as they explored the interface (but could turn their videos off) and the researcher recorded the screen using Zoom's "record meeting" feature. The researcher also took notes on any misinterpretations of the data by the participants, usability issues, and other salient exploratory behaviors of the participants. The researcher prompted the participants to speak aloud if they did not do so as they explored the interface.

After the participants finished exploring the interface, they were asked a couple of follow-up questions on their perceived usefulness of the types of data shown on the interface. Then the researcher stopped the recording. Finally, they were asked to fill out a short questionnaire (with screen-sharing off) on user-engagement [38], which included the 3-item **Aesthetic appeal** subscale measuring the attractiveness and visual appeal of the interface. At the end of the session, the researcher answered any questions the participants had and each participant received $25 for their participation. Following the Zoom sessions, the researcher also emailed the participants the unique links to their respective visualizations for them to view at leisure. The questionnaires, instructions, and additional results are included as part of supplementary material.

### 4.3 Coding Process

We first transcribed the audio recordings from the recorded Zoom sessions using the *Otter.ai*[2] transcription service. We then manually checked the generated transcripts and added additional visual evidence information (e.g., the feature type clicked) within square brackets to accompany participant observations by going over the screen-capture recordings. We then segmented the transcripts and defined our unit of analysis similar to Choe et al. [13] as an individual observation about the data shown on the visualization interface.

For the coding process, we adopted the insight types identified in [13, Table. 1] and [14, Table. 2] as predefined codes and viewed the screen-capture recordings for each participant and applied the appropriate code(s) to the individual transcript segments. In addition, to answer **RQ3**, we coded any (i) misinterpretations of the data by the participants, (ii) questions that participants had about the interface or the data shown in the interface, and (iii) user-interface design factors hindering effective interpretation and exploration of the data. The first author performed the entire coding process.

### 4.4 Interaction-Log Metrics

The interface automatically logged interactions of each participant during the think-aloud sessions and using these logs, we computed their exploration time, the time they spent reading the text descriptions, the number of times they clicked on summary types and feature types, and the number of times they hovered on data points

[1] https://zoom.us/

[2] https://otter.ai/

in the *You_graph* and *Baseline_graphs*. We computed the time spent in reading the text descriptions by computing the times when the mouse was in the text description area and we corroborated these times from the participant videos.

## 5 RESULTS

We present the identified insight types (**RQ1**), context types (**RQ2**), and potential barriers to insight generation (**RQ3**) from coding participant verbal reports below. Summary statistics of and Spearman's correlations among the questionnaire responses (Aesthetic appeal and Data savviness) and interaction log metrics are presented in Tables 1 and 2, respectively.

### 5.1 Insight Types

Participants' exploration time ranged from 4.13 to 52.33 minutes with a median exploration time of 8.37 minutes. We gathered a total of 203.48 minutes of think-aloud data from all the participant sessions and extracted 278 personal insight observations from this data (median = 15, min = 6 (P4), max = 40 (P3)). From these observations, we identified 797 insights (avg ≈ 3.0/observation).

Table 3 presents the insight types. We revised the definition of the "Comparison against external data" insight type presented in prior work [13, 14] to "Comparison with baselines" to fit the design of our interface. We did not include the "Comparison with multiple services" defined in previous work [14] because it was not applicable to our interface.

While the insight types identified in prior work are applicable to those found in our study, there are differences in the relative frequency identified for each type. The insights gained by participants are influenced by the design of the interface and hence, we suspect that the interaction of hovering over data points on our interface may have led to the high frequency of the "Detail" type.

### 5.2 Context Types

To answer **RQ2**, we looked at the transcript segments (83) containing the "External context" insight type (see Table 3) to identify the types of contextual information recalled by participants to interpret their data. We categorize the identified contextual data into the following five types (frequency for each type are included within parentheses).

#### 5.2.1 Specific Personal Events Occurring During the Tesserae Study Period (26)

Participants recalled various events that occurred during the time frame of the Tesserae study to interpret their data shown in the interface. Of these events, mentions of travel or vacations were the most significant (11). For example, one participant mentioned, *"I'm looking at the steps. I'm looking at the average daily steps per week. And just looking at the times, the drops that I had, and then thinking about what went on that week, like the week of July 29th, I only averaged 5000 steps, but we were in a plane. We traveled that week. So I know two of the days was spent sitting in an airplane for four and a half hours, so that would definitely cause my step counts to go down" (P7).*

Participants also recalled being sick, their work schedules, and other personal events that occurred at specific times over the course of the Tesserae study. For example, one participant stated, *"My office hours–Oh my goodness! So I definitely peak whenever we come back to budgets in the Fall, August and September. I really start getting busy. I think put in extra time because of that" (P5).* In accounting for their home and office hours at certain times, another participant commented, *"Yeah, interesting. I'm remembering that these are times that I was involved in musical theater and was away a lot for rehearsals. So, not at work but also not much at home either" (P6).*

#### 5.2.2 General Activities or Habits of Participants (19)

Participants also stated their general activities or habits to make sense of the data they were seeing. These activities and habits are not specific to the time frame of the Tesserae study and are comparable even during the present time, such as, their sleep or physical activity routines. For example, one participant commented, *"My sleep is generally about seven hours pretty consistently. It's funny that it's showing this. I don't have sleep apnea or anything like that. I don't wake up at night" (P3)* and another said, *"I would expect myself to be probably above the cohort and all people because I am a pretty active person" (P10).*

We observed that participants also stated their self-tracking goals as they interpreted their data. For example, a participant stated, *"Also makes me think about, like, those kind of things like, if I'm spending a lot of time in the office, I need to make sure I'm getting.. I'm active at home, since I'm not as active as I can be at work, to try to balance it all out. And that I used.. like steps was always.. I mean, I've worn a fitness tracker for a lot of years and steps was always my biggest thing, making sure I got 10,000 or 11,000 a week, but in recent years, it's been also making sure I'm getting the right amount of sleep" (P7).*

#### 5.2.3 Study-Related Information (17)

Participants also recalled various aspects of the Tesserae study including the start and end dates of their participation, other participants in the study, the types of data gathered, the devices used to gather data, and the times when they forgot to wear the Garmin wearable or when the devices failed to function correctly. For example, a participant attributed data that contradicted their expectation to potential issues with their wearable, *"Just sort of surprised that my steps are high but my activity is not. That's where I was having an issue with the Garmin I think not recording activity very well. They still do that at times" (P10).*

Participants remembered the requirements they completed as part of the study to explain their data–*"I do remember we took those quizzes frequently that asked all these questions about how stressed we were, how tired we were, and how we felt about everything. So I generally didn't feel very stressed on those. So I guess that's good that my number is high" (P3)* and *"[I'm] at home a lot more than the rest of the individuals. I just remember that we actually had a tracker - one in our home and office. So looking at the office hours, I could see why that would be a lot different" (P15).*

The baseline ("cohort" and "all") graphs shown on the interface were also closely observed. For example, one participant observed the sudden decrease in the number of "all" participants (encoded using a sequential color scale) between two consecutive time periods, *"I wonder why it dropped from 735 to 648 between those two points, is that normal? 735, 735, 735. It looks like 735 pretty consistently. So at some point, there must have been, I guess right here between January, a study must have ended with some cohort because it totally drops a bunch of people right there" (P9).*

#### 5.2.4 General Holidays (12)

Participants very quickly recognized certain data points possessing extreme values because they were gathered over the Christmas or Thanksgiving holidays. For example, one participant reminisced, *"I think the heart rate one was very interesting to me. That I was very relaxed during the Christmas break. Also, I think that was around the time the winter classic was happening and I was very relaxed and having a good time with my dad and son" (P5).* Participants also noted variations in their data explained by weekends–*"I'm guessing the high points [in home hours] are the weekends. That would make a lot of sense" (P2).*

Table 1: Spearman's correlations of questionnaire responses (*Aesthetic appeal* and *Data savviness*) and interaction log metrics of the 15 participants. (*** = p<0.001; ** = p<0.01; * = p<0.05).

|  | Aesthetic appeal | Data savviness | Exploration time | Reading Time | # of *Summary* clicks | # of *Feature* clicks | # of *You_graph* hovers |
|---|---|---|---|---|---|---|---|
| Data savviness | -0.38 | | | | | | |
| Exploration time | **-0.59 *** | **0.55 *** | | | | | |
| Reading time | -0.50 | 0.43 | 0.43 | | | | |
| # of *Summary* clicks | -0.21 | **0.58 *** | 0.38 | 0.15 | | | |
| # of *Feature* clicks | -0.16 | 0.31 | **0.55 *** | **0.56 *** | 0.12 | | |
| # of *You_graph* hovers | -0.49 | 0.39 | **0.74 **** | 0.50 | 0.17 | **0.60 *** | |
| # of *Baseline_graph* hovers | -0.46 | 0.35 | **0.64 **** | 0.49 | 0.23 | **0.64 **** | **0.96 ***** |

Table 2: **Range** (p.r. denotes possible range) **and median of the variables across all 15 participants.**

|  | Range | Median |
|---|---|---|
| Aesthetic appeal | [2, 4.33] (p.r. [1, 5]) | 3.67 |
| Data savviness | [1, 4] | 3 |
| Exploration time | [4.13 min, 52.33 min] | 8.37 min |
| Reading time | [0, 337 seconds] | 18 seconds |
| # of *Summary* clicks | [0, 55] | 9 |
| # of *Feature* clicks | [7, 159] | 17 |
| # of *You_graph* hovers | [0, 563] | 80 |
| # of *Baseline_graph* hovers | [0, 790] | 137 |

#### 5.2.5 Seasonal Patterns or Trends (9)

Participants also recalled seasonal patterns and trends to interpret the data shown in the interface. For example, one participant attributed the decrease in activity to seasonal changes, *"Summer time and Fall being out and about versus wintertime where we get a little bit more dormant" (P13)* and another observed sleep changes across the board, *"It's interesting to see during the summer months, sleep averages drop a little bit, just because there's more Daylight Time. It's harder to get the cues that it's time to go to bed if the sun still shining" (P7).*

### 5.3 Shortcomings in the Interface Design

To answer **RQ3**, we categorized participant observations coded as misinterpretations and interface design issues into the following five types.

#### 5.3.1 Potentially Misleading Baselines

The "cohort" and "all" baselines visualized on the interface represent the *average* values for all the participants in the cohort and the study, respectively, and hence, they appear smoother and less variable compared to a participant's weekly data superimposed on the baseline graphs. This difference led ten out of the 15 participants to interpret their (weekly) data as being inconsistent and "all over the place" compared to the baselines. For example, one participant stated, *"It's interesting that out of all the options listed, mine is sporadic compared to the other two. It's crazy. I don't really know what that means" (P4).*

In addition, we also observed that participants generally overly relied on the baselines to interpret their personal data. They viewed baselines as representing ideal values and formed opinions about themselves based on how their data compared to the baselines. For example, one participant commented, *"[Monthly activity graph] Yeah so I did pretty good in May, did pretty bad in July ..did pretty good in October ..huh yeah I guess I was a very bad study participant because I mean the trend [baselines] is here and I'm just like way down here like you shouldn't have way down here it should be similar." (P9).*

#### 5.3.2 Less Clear Data and Textual Descriptions

Three participants missed reading the definition for "heart rate variability (HRV)" (see Section 3.2.1) and misinterpreted it as "heart rate" and two participants commented that HRV was too technical to interpret. One of them said, *"Yeah I'm just trying to figure out what it is answering for me. If the answer was how stressed was I feeling, the data is too confusing for me as an average person to interpret. So I am actually not very sure what it is telling me. I know how it's defining what it is telling me but my brain isn't connecting it with a useful piece of information" (P3).* In addition, three participants were not clear on how the "activity" feature was defined and how it differed from "steps" and three participants wanted to know more details about the "cohort" and who were included in that group. For example, one participant commented, *"Who's the Cohort that they spent six and a half hours-ish in the office. Depends on what their office is. Can't assume that they do the sort of job that I do" (P3).*

#### 5.3.3 Insufficient (Tesserae) Study Context

While participants recalled various aspects of the Tesserae study as they explored their visualizations (see Section 5.2.3), we observed that the study-related information provided on the interface was inadequate and presented potential barriers to effective interpretation of the data. While some participants were able to recall the reasons for the missing data they noticed on the interface, four participants were surprised to see missing data. For example, one participant said, *"So the office hours.. man, I know I kept one of those trackers at the office. Does my data just not show? Was my data not collected?" (P13).* Further, one participant incorrectly attributed their missing "office hours" data to potential issues with their wearable when that information was actually computed using their beacon–*"[Office hours] So I assume that number of days is the number of days that got recorded. So like the week of July, they got me five days in the office. But this one, only four. So my wearable must have died" (P9).*

#### 5.3.4 Potentially Ineffective Visual Encodings

We observed that participants often missed noticing certain types of information or that certain information was not communicated to them clearly. For example, two participants found it difficult to discern the number of participants corresponding to each data point in the baseline graphs (visually encoded using sequential color scales) by comparing the color hues. One of them said, *"Yeah, I don't think that the shading is that helpful in determining the number of participants because there's not enough level of detail in the shading for me to tell like how many participants there were... I mean, like this difference between 735 and 648 is impossible for me to tell on this chart and that's a significant drop in how many people" (P9)* and the other found it easier to discern the differences on the weekly graphs compared to the monthly graphs, *" I don't notice the color differences as much in the monthly scale, the variation between like the reds and the greens but in the weekly, you can clearly see where it was a definitely heavier participant count versus a lighter participant count" (P7).* Additionally, five participants did

| Type (total frequency) | Subtype (frequency) | Description | Example Quotes |
|---|---|---|---|
| Detail (337) | Identify references (124) | Explicitly state the values of categorical variables, labels from the axes, or legends | *"So now I'm looking at the number of participants in the cohort" (P10)* |
| | Identify value (122) | Explicitly specify the measured value, its range for one or more clearly identified data points, or the difference between two measured values | *"Office hours. OK April eight and a half hours" (P3)* |
| | Identify extreme (91) | Explicitly state the identities of the data points possessing extreme values of the measure variable | *"That's interesting.. A two-hour day at the office. That's kind of weird" (P11)* |
| Recall (164) | External context (83) | Uncaptured data provided by the self-tracker to understand and explain a phenomenon shown in the data | *"This must have been spring break. Think we went to Florida that year" (P3)* |
| | Confirmation (57) | Collected data confirms existing knowledge | *"Let's do this home hours. Yeah, I'm a homebody alright (P12)"* |
| | Contradiction (24) | Collected data contradicts existing knowledge | *"Yeah, I guess I'm mostly more active than cohort and the all which is somewhat surprising to me. I don't feel like I'm all that active per se" (P1)* |
| Comparison (136) | With Baselines (75) (specific version of *Against external data* in [13, 14]) | Compare with baselines presented on the interface | *"So it looks like I'm more relaxed than everyone in the cohort" (P10)* |
| | By Time Segmentation (37) | Compare measured values segmented by time | *"[Activity] Yeah so I did pretty good in May, did pretty bad in July.. did pretty good in October" (P9)* |
| | By factor (13) | Compare measured values by a factor (other than time) | *"It's interesting to see a dip in the average HRV during the holiday weeks, like the week of December 23, there was a dip and then around Thanksgiving, there was one too-a slight dip" (P7)* |
| | Instances (11) | Compare two specific instances | *"So seven and half hours I guess at home. And Almost 8 hours in the office. So where are the other ten hours?" (P3)* |
| Data summary (56) | | Summary of collected data (such as number of data points, duration of tracking, and averages) | *"Christmas break was good for me for sleep. In those two weeks, I averaged over nine hours" (P7)* |
| Trend (32) | | Describe changes over time | *"I did OK in February but then I went back down a little bit and stayed down" (P5)* |
| Distribution (29) | Variability (29) | Explicitly state the variability of measured values | *"Ok, so this shows I guess that I have had some very big peaks and valleys compared to everyone else" (P2)* |
| | By Category (0) | Explicitly describe the variation of measured values across all or most of the values of a categorical variable | – |
| Value judgment (25) | | Convey positive or negative connotations about the data | *"Man, look at me champion sleeper in September. Really, really did good for one month" (P9)* |
| Correlation (11) | | Specify the direct relationship between two variables (but not as comparison) | *"Typically, if it was an active week, I typically had good sleep." (P7)* |
| Prediction (4) | | Predict the future based on the collected data | *"It'll be interesting to see if you're more active at home versus office... My assumption would be that we're more active at home than we are in the office" (P7)* |
| Outlier (3) | | Explicitly point out outliers or state the effect of outliers | *"Yeah, I'm on average except for a couple of outliers" (P12)* |
| **Total (797)** | | | |

Table 3: Types and descriptions of visualization insights drawn from [13, Table. 1] and [14, Table. 2] with example quotes and frequency from our study.

not notice missing data points as they interpreted the trends in their line graphs and four participants missed noticing the low "number of days" (which is displayed when they hover over the data point) over which the value reported in the data point was gathered, potentially leading to misinterpretations of the data in both cases.

### 5.3.5 Inadequate Interaction Affordances

Two participants were particularly interested in exploring the correlations and relationships among the feature types. One of them clicked through all the feature types to be able to relate the corresponding feature values for individual data points; they said, *"So it's interesting to see how everything comes together and works its way, like out within yourself physically. Like if you're low active, you may have difficult sleep that time, your heart rate might be showing that you're a little bit more stressed. And you could be possibly spending more time in the office and your step count goes down. So it's interesting to see how it's all interrelated" (P7).* The other participant opened the visualization link in two browser tabs and split their screen during the study to be able to view pairs of feature types side by side.

Participants also commented on additional interactions they wished they could perform on the interface. For example, one participant said, *"It's great to see my data against the cohort and the all but it would be kind of cool if I could like turn off the all folks and cohort and just look at me ..my data for a bit" (P3).* Another participant wanted an easier way to be able to switch back and forth between the monthly and weekly summaries for each feature type and also said it would be helpful if they could adjust the range on the y-axis so they can see the values of the data points more clearly– *"[Monthly sleep graph] I'm all pretty much very close to the cohort. So there's not really much span it looks like. It might have been more discernible to be able to read it if the number of hours duration.. because there is none from zero to six.. if maybe it would have been spread between six and eight or nine, it would have been easier to see the distinction between myself and the cohort and everyone" (P10).*

## 6 DISCUSSION

We discuss our findings and possible design directions for enhancing visualizations in self-tracking apps below.

### 6.1 Embedding Contextual Cues

Li et al. report *context* as one of the six questions that people ask about their data in order to self reflect [30]. Embedding additional context information can help trigger people's memory and enable them to more effectively interpret their data and gain richer insights [6, 24, 31]. We aimed to define the context information recalled by participants in our study (**RQ2**) and present five context types in Section 5.2.

The most frequent types (types (i) and (ii)) represent information that is specific to individuals, such as, their personal experiences, travel, routines, activities, and social interactions [24]. Types (i) and (ii) can inform self-tracking apps of additional information that they can prompt users to manually log or annotate (similar to *TimeMarks* [23]) via the apps. Apps can also be designed to enable users to extract and embed such context information usually logged using other means (for example, from calendars and journals) [5, 39]. Additionally, including more general context information (types (iv) and (v)), such as, weekends, major holidays, and seasons, within personal visualizations can be useful. The context information can be embedded directly in the chart or made to appear when interacted with the corresponding data or even as an additional textual "narrative" element on the side linking the text and corresponding data.

### 6.2 Data Savviness, Exploration Time, and Aesthetics

We found Spearman correlations (see Table 1) of 0.55 ($p < 0.05$) between *Data savviness* and *Exploration time*, 0.58 ($p < 0.05$) between *Data savviness* and *# of Summary clicks*, and -0.59 ($p < 0.05$) between *Exploration time* and *Aesthetic appeal*. The correlations suggest that participants who were more data-savvy spent more time in exploring the interface and also performed more overview+detail interactions (or *Summary* clicks); and participants who spent more time rated the interface as being relatively less aesthetically appealing.

Following recommendations from prior work for evaluating personal informatics systems [26], we designed our interface to be relatively simple so that we could directly tie our findings to the interface design aspects. Hence, it is not surprising that participants, especially those who were more data savvy, identified limitations in the interaction mechanisms included on the interface (see 5.3.5). Participant comments in 5.3.5 allude to the need for options for filtering, faceting, zooming to switch between overview and detail views [35](Ch. 11-13), and axis rescaling [42] on the interface. Further, we expect that these participants are accustomed to using the more visually-appealing commercial tools available which could explain their relatively low rating of the aesthetics of our interface. Our findings are in line with prior work [16] that suggests that personal visualizations targeted at more data-savvy users should have higher *explorability* (i.e. the ability to explore data on the interface [24]) and that aesthetically appealing visualizations can increase user engagement [5, 24, 49].

### 6.3 Defining Baselines for Comparison

People gain insights by making comparisons [13]. However, defining suitable baselines for comparison is one of the key challenges that arise when designing personal visualizations and any baseline included will likely bias a user's interpretation of their data [24]. Large-scale sensing studies, such as the Tesserae study, provide opportunities for defining baselines for comparison because they gather the same types of data somewhat consistently from specific target populations. Participants in our study interacted quite a bit with (see Table 2) and were generally appreciative of the baseline graphs, especially because apps associated with wearables such as Fitbit and Garmin generally do not provide such baselines. For example, one participant said, *"I think using the interface was very helpful for me to compare to others in the study" (P5).*

However, we also found that the presented baselines were potentially misleading (see 5.3.1) and that participants failed to interpret data aggregations correctly. It might be more effective to consider alternative representations of baselines that do not give rise to the issue of *dissimilar variability*. Our findings also suggest that users may want the option to look at their data in isolation as well, without the baselines.

### 6.4 Data and Textual Descriptions

Some of the features presented (for example, HRV) are *derived* features computed using the raw data from the Tesserae study and hence, considering participants' potential limited data and/or visualization literacy, we included detailed textual descriptions alongside each visualization describing the data as well as the visualization components. Including explanatory text is a common practice in visualization (also employed by the New York Times, e.g., [36]), especially when the goal is to convey the data accurately to many audiences and reduce misinterpretations of the data [50].

However, we found that, explanatory text and unclear data presented potential barriers to interpretation regardless of participants' exploration time and interest. Participants who spent little or no time in reading the descriptions misinterpreted some of the data and participants who spent more time in exploring the interface found the descriptions to be both lacking in clarity and wordy (see 5.3.2).

For example, one participant commented *"The right side of the interface was too text-heavy" (P3)*. The HRV feature was especially confusing to participants; one of them suggested, *"Simply communicate how stressed you are and not stressed you are" (P10)*. Hence, our findings suggest that, in addition to presenting clear, unambiguous data, it might be effective to minimize the need for reading in general on personal visualization interfaces and to achieve higher explorability through additional interaction mechanisms rather than more textual descriptions. Additionally, comprehension can be facilitated through the use of visualization components, such as, titles, axes labels, legends, and annotations [8, 17].

## 7 LIMITATIONS

Through the work presented in this paper, we contribute to the prevalent but less-researched area of personal visualizations. However, our study has some limitations. Our findings are influenced by our design decisions for the interface, including the use of line graphs and choice of visual encodings and data presented on the interface. Further, our interface may not be representative of contemporary personal informatics tools that provide a variety of visualizations as well as offer options for customization in terms of interface layout, aesthetics, and other factors [11, 41, 44]. We also acknowledge that the contextual cues reported from our study (in Section 5.2), while useful, may also be incomplete and it is likely that participants would have remembered additional and/or different information had the study been conducted soon after the Tesserae study ended.

There may be potential bias with using existing codes/insight types from prior work in our coding process to categorize the insights gathered in our study. However, our main aim with this approach was to see if the codes were also applicable in our case and to use them as a means to evaluate the interface's ability to support insight generation.

## 8 CONCLUSIONS

We have explored some of the design challenges in personal visualizations through a mixed-methods study of visualizations of the personal data gathered from the Tesserae study [34]. Given that quantified selfers routinely review their past data [13, 19], we identified types of contextual cues that can supplement the *quantitative* data from wearables and sensors and enable users to *subjectively* interpret their data. In contrast to prior studies in personal visualizations that have mainly targeted users who are more motivated and/or are existing personal visualization users [26], we studied participants with varied levels of data-savviness and how this difference affects their exploratory behaviors. Finally, our findings highlight the importance of aesthetics, clear data, and limited text in personal visualizations.

### ACKNOWLEDGMENTS

This material is based upon work supported by the National Science Foundation under Grant No. SES-1928645, SES-1928718, SES1928612, and SES-2030599.

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
