# OpenReview forum: "Triggers and Barriers to Insight Generation in Personal Visualizations"
_graphicsinterface.org/Graphics_Interface/2022/Conference — GI 2022_

### Official Review · Reviewer_rjvr · 2022-01-05
**Triggers and Barriers to Insight Generation in Personal Visualizations**

**Rating:** 6
**Confidence:** 3

**Review:**


This submission presents findings from a think-aloud study with 15 participants who were asked to review their personal sensor data, which was collected via a separate study that took place approximately 1 year prior.  This current study’s focus is on a newly designed visualization interface; the original data collection study is described in a referenced publication. Findings from the think-aloud study describe insights participants were able to glean from their data as well as challenges they faced with the current visualization interface design.

Strengths

The authors have a rich dataset to work with.  There is a lot of potential to use this dataset to explore a range of visualizations.  In future work, it would be interesting to see a deeper, more systematic exploration of the design space.
This current study seems like it was well run.  In terms of findings, I found two results interesting and potentially valuable for designers.  The first was the how the interface’s comparisons to averaged baselines made participants question why their data was so jumpy.  It would strengthen the discussion section to have more reflections or ideas on what designers might do to mitigate this issue.  I also found the finding related to having more context or landmarks available interesting as participants seemed to want to recall this context when reflecting on their data.  One caveat is that participants were reflecting on data collected over a year ago and so it isn’t clear whether this finding might generally to short-term use cases, which I suspect are more common.   Other findings felt more specific to usability issues with this visualization and so were less compelling from a scientific standpoint.

Weaknesses

The paper describes the study task as free exploration. While a non-prescribed study task can be fine, I found it difficult to interpret the findings without knowing what tasks participants wanted to accomplish with the visualizations.  Visualizations are typically designed to support certain tasks.  What tasks were the participants creating for themselves?

As mentioned above, participants were asked to reflect on data collected over one year earlier.  Is this a common use case for personal data visualizations?  I would like to see further discussion of this study design decision and how it impacts the generalizability of the study findings.

The paper presents a finding about data-savviness and how those users wanted more sophisticated manipulations.  There is some depth of analysis missing here to really get at the underlying issue.    Is this main issue that these users have greater expectations built up from experiences with other tools, or is there something else at play?

The thing that I struggled with the most with this submission is assessing the novelty and strengths of the contribution. In terms of comparisons to prior work, there is some discussion of this being an underexplored design space, but I found these statements too vague to contextualize the work.  The paper could do a much better job of establishing the novel aspects of the visualization interface and how the study findings extend current knowledge.  The paper indicates that a main novel feature is how the data was collected, but there are numerous examples of personal data tools that include longitudinal observations for individuals and groups (e.g., Strava comes immediately to mind as a deployed example).

In terms of the size of the contribution, part of the issue is a non-standard introduction that largely skips the important step of establishing and motivating the problem that the paper is trying to solve.  The introduction immediately opens with a paragraph comparing to related work, without first establishing and motivating the problem the paper is contributing knowledge towards.

Overall Assessment

Weighing the strengths and weaknesses, I lean slightly towards recommending “Accept”.  The paper presents a couple of nice findings, but there are also some missing pieces.  The submission would benefit from a much stronger motivation and comparison to prior work.  It is missing justification on the value of supporting this kind of longitudinal personal data reflection.  The interface design is also lacking a clear discussion of novelty.

---

### Official Review · Reviewer_YEKF · 2022-01-13
**A few interesting observations about context, but too many unjustified generalizations for a very limited study.**

**Rating:** 6
**Confidence:** 4

**Review:**

This submission describes findings and reflections derived from a set of 15 think-aloud interviews focusing on a visualization tool developed as part of a large-scale personal informatics study. The authors detail related work in personal informatics and visualization, provide a brief overview of the Tesserae Personal Data Explorer and their study, and describe observations derived from the think-aloud transcripts.

The submission is fairly clear and provides a reasonable overview of related work. However, the scope and depth of the study are limited, and, as a result, the analysis and observations aren't really able to provide answers to the extremely general research questions the paper proposes to answer. This, plus the relatively limited and specific nature of the visualizations developed for this system, make it unclear to what extent most of the observations reported in the paper could apply to other personal informatics tools or experiences.

In general, the functionality supported by the personal data explorer system is extremely simple and not very representative of most contemporary personal informatics tools. The main novelty in the tool comes in the form of baseline data from other participants in the larger Tesserae study. However, the way this data (and information about how to interpret it) is surfaced and explained in the interface seems somewhat confusing – relying on color encodings that seem challenging to interpret in the visualizations and large blocks of dense text alongside them. As a result, many of the observations (problems with misleading encodings, unclear text, etc.) seem to reflect issues with this design and it isn't clear to what extent they provide actionable insights for the design of other systems.

The way the study was conducted - presenting participants with aggregated visualizations of their data out of context and close to a year after data collection - also seems out of sync with most personal informatics tools. Again, many of participants' experiences with the visualizations seem likely to have been heavily impacted by the fact that they were viewing this data so far after collection that it was difficult to connect to events, calendars, and other context.

Overall, I find that the results aligning with RQ2 ("What are the types of contextual information recalled by participants?") are the most compelling, and I can imagine how the list of context types presented in Sections 5.2 and 6.1 could serve as a useful starting point for the designers of other kinds of personal visualizations systems. (Although, I do expect that the kinds of observations that participants made were heavily impacted by the delay between data collection and the think-aloud study, so such a list is likely incomplete.) On the other hand, RQ1 ("What are the insights that participants draw from visualizations of their personal data?") and RQ3 ("What interface design choices are potentially hindering insight generation?") are never answered in a way that seems likely to generalize beyond this particular tool.

As a result, the possible design directions discussed at the end of the paper feel underwhelming and problematic. I very much like the idea of embedding contextual cues (Sec 6.1) and think this is the most positive outcome of the submission. However, the points about explorability and aesthetics in Sec 6.2 seem well-established in existing practice (most PI tools already do some of this). Meanwhile the concerns about defining baselines and including text descriptions seem very context-dependent. While the specific ways baselines and text explanations were implemented in this tool clearly caused issues, the kinds of general takeaways ("don't show baselines by default", "minimize reading") given in 6.3 and 6.4 seem like *colossal* overstatements made with far too little evidence.

In summary, this study represents a reasonable (but extremely limited) evaluation of the Tesserae Personal Data Explorer. It contains some interesting observations and a list of external context types that might serve as a useful starting point for others seeking to incorporate contextual references into personal visualizations. However most of the rest of the observations feel like they reflect usability issues with the current system, and the generalizations made based on them are unwarranted. While I'm okay seeing most of the paper published and available as part of the literature, **I think the claims in 6.2, 6.3, and 6.4 need to be *dramatically* softened or even eliminated.**

---

### Official Review · Reviewer_euHF · 2022-01-15
**Small but valuable contribution to the area of personal visualization**

**Rating:** 7
**Confidence:** 4

**Review:**

This is an interesting piece of research that relies on a large, real-world dataset of personal data. This is a small piece of a large scale research project that looks at some questions specific to personal visualization. I particularly appreciate the effort to maximize ecological validity of the study, with participants looking at their own personal data collected for over a year using a personal visualization digital prototype.

One aspect that is a bit overlooked in the related work and discussion of the results is the effect of the particular interface on the results. More specifically, previous research has shown that customization and preferences in terms of interface layout, aesthetics, and so on, are both subjective and important when it comes to personal visualization, with many personal visualizations going quite beyond the standard visualization idioms such as the one presented in this prototype (a line graph) - see [A, B, C] for example. This is an aspect that is not supported in the prototype and that I would expect to see being part of the discussion.
That being said, the simple/standard interface provided is also reflecting very well what exists in the real world and what people are already using (e.g., Garmin, Fitbit, or Nike apps all rely on these similar standard charts).

The study is well designed and appropriate to answer the research questions (using an insight-based evaluation while collecting a combination of qualitative and quantitative information). I also thank the authors for providing the study data in supplemental material.
What is missing from the supplemental material though, is the list of codes/insight types that were used in the coding process. The paper points to tables from other research papers, which I do not think is fully appropriate. I strongly encourage the authors to include the full list of insight types they used in supplemental material (it is unclear from the paper alone if the insights in Table 3 are the union of insight types drawn from the two references listed, or the intersection, or something else; it is also unclear whether these two references do provide insight types that had 0 occurrences in the presented study).
The coding process and data analysis are appropriate. I would like a bit more details about the coding process, such as how many coders were involved, what the different stages of the process were, was any inter-coder reliability metric used, and if not, why, etc..
I also find a bit concerning that some participants performed 0 clicks, 0 hovers, had 0 reading time, or spent 4 minutes only on the visualization. Because of this, the absolute frequency of insights is not completely meaningful (or at least not enough to get the full picture) because there is a huge disparity in terms of number of insight and 'involvement with the experiment' from participants. Ways of addressing this would include providing a summary table with the detailed data for each individual participant (instead of summary statistics provided in Table 2); or even better, to complement Table 3 with a mini-visualization that would visually convey the number of insights of each type for each participant (i.e. using some kind of tabular visualization). Worst case would be to provide this information in supplemental material, but from my perspective this is crucial information to fully understand the data and should be included in the paper in a condensed form.

Overall, the results of the study are interesting, though not earth-shattering. They largely confirm previous findings, but with small variations that are worth disseminating. This is an incremental piece of research, with a relatively small contribution, but still a valuable one as some of the suggestions provided in the discussion can help future researchers/app designers create personal visualization interfaces.


[A] F. Rajabiyazdi, C. Perin, L. Oehlberg, and S. Carpendale. Exploring the design of patient-generated data visualizations. Graphics Interface 2020.
[B] A. Thudt, C. Perin, W. Willett, and S. Carpendale. Subjectivity in personal storytelling with visualization. Information Design Journal, 2017.
[C] J. Rodgers and L. Bartram. Exploring Ambient and Artistic Visualization for Residential Energy Use Feedback. IEEE TVCG, 2011.

---

### Decision · Program_Chairs · 2022-01-18

Accept